# Combination of Cannabidiol with Cisplatin or Paclitaxel Analysis Using the Chou–Talalay Method and Chemo-Sensitization Evaluation in Platinum-Resistant Ovarian Cancer Cells

**DOI:** 10.3390/biomedicines13020520

**Published:** 2025-02-19

**Authors:** Jana Ismail, Wassim Shebaby, Shirine Azar Atallah, Robin I. Taleb, Sara Kawrani, Wissam Faour, Mohamad Mroueh

**Affiliations:** 1Pharmaceutical Sciences Department, School of Pharmacy, Lebanese American University, Byblos 1102 2801, Lebanon; jana.ismail@uni-jena.de (J.I.); wassim.shebaby@lau.edu.lb (W.S.);; 2Department of Natural Sciences, School of Arts and Sciences, Lebanese American University, Byblos 1102 2801, Lebanon; robin.taleb@lau.edu.lb (R.I.T.);; 3Gilbert and Rose-Marie Chagoury School of Medicine, Lebanese American University, Byblos 1102 2801, Lebanon; wissam.faour@lau.edu.lb

**Keywords:** cannabis, CBD, ovarian cancer, cisplatin, paclitaxel, synergism, antagonism, sensitization

## Abstract

**Background/Objectives:** Cannabidiol (CBD) is known for its anti-cancer properties in preclinical models and is increasingly used alongside conventional chemotherapy in cancer treatment. This study aims to evaluate the anti-cancer activity of CBD from Lebanese *Cannabis* sativa as a monotherapy and in combination with cisplatin or paclitaxel on human ovarian adenocarcinoma cells. **Methods**: Cytotoxicity of CBD was tested on OVCAR-3 and SK-OV-3 cell lines using the MTS assay. The Chou–Talalay method and CompuSyn software were used to determine the combination indices (CIs) for predicting interactions between CBD and chemotherapeutic agents. CBD showed dose-dependent tumor growth inhibition at 72 h with comparable IC_50_ values for both cell lines. **Results**: The combination of CBD with cisplatin or paclitaxel showed significant antagonistic interaction in SK-OV-3 cells (CI > 1), but mild synergism (CI < 1) at high growth inhibition rates (95% and 97%) was observed in SK-OV-3 cells with CBD/cisplatin. Pure antagonism was found in OVCAR-3 cells with CBD/cisplatin. Priming SK-OV-3 cells with CBD reduced the IC_50_ values of both drugs significantly, with a similar effect seen when cells were primed with cisplatin or paclitaxel before CBD treatment. **Conclusions**: Integrating CBD with chemotherapy could improve cancer therapy and address drug resistance. Sequential administration of CBD and chemotherapeutic agents is more beneficial than simultaneous administration. Further in vivo studies are necessary to validate these findings and understand CBD’s interactions with other drugs fully.

## 1. Introduction

*Cannabis* is a complex system comprised of various pharmacologically active classes of compounds, such as cannabinoids, terpenoids, alkaloids, and flavonoids. Among the 120 cannabinoids identified, Cannabidiol (CBD) and Tetrahydrocannabinol (THC) are the most commonly recognized, as they are present in the highest abundance in the *Cannabis* plant and have been extensively investigated since their discovery [1]. However, unlike THC, CBD has no reported psychoactive effects and thus presents as an attractive cannabinoid to study.

CBD has been reported to possess anxiolytic, anti-psychotic, anti-inflammatory, and anti-hyperalgesic properties [2,3,4]. Recently, a CBD oral solution (Epidiolex^®^) has been approved by the US Food and Drug Administration (FDA) for the treatment of certain seizure syndromes [5]. CBD in vitro monotherapy has also been extensively studied against various cancer cell lines [6]. In addition, several studies have been conducted to investigate the anti-proliferative effect of CBD in combination with chemotherapeutic agents. The reported literature varied from notable synergism to additive effect, and to prominent antagonism, depending on the adjunct drug administered with CBD, as well as the specific cancer cell line tested [7,8,9,10,11,12,13,14,15,16,17,18,19,20,21].

Ovarian cancer is one of the most lethal, aggressive, and resistant cancer types among women [22]. Current treatment approach consists of surgery and chemotherapy. Chemotherapeutic regimens, consisting of a combined platinum drug and a taxane, are associated with debilitating side effects and rapid development of resistance. Cisplatin and paclitaxel are two of the most prescribed therapeutic regimens, and reports have shown an increasing trend in treatment failure due to drug resistance [23]. This calls for research on novel effective and safer alternative therapeutics capable of combating chemotherapy resistance.

In 1986, the FDA approved THC (Dronabinol^®^ and Nabilone^®^) for the treatment of chemotherapy-induced nausea and vomiting (CINV). Moreover, current clinical trials are investigating the combination of THC and CBD for the same indication [24,25]. However, the effect of combining CBD and chemotherapy has not been well investigated. Hence, this study aims to investigate the anti-cancer potential of CBD on platinum-resistant ovarian cancer cell lines OVCAR-3 and SK-OV-3, alone and in combination cisplatin or paclitaxel, and evaluate the potential of combating chemotherapy resistance.

## 2. Materials and Methods

### 2.1. Reagents

Human epithelial ovarian cancer cell lines SK-OV-3 and OVCAR-3 were purchased from American Type Culture Collection (ATCC, Gaithersburg, MD, USA). Dulbecco’s modified Eagle’s medium (DMEM), RPMI 1640 culture medium, bovine insulin solution (sterile-filtered) as well as GCMS solvents methanol, dimethyl sulfoxide (DMSO), dichloromethane (DCM), and ethyl acetate were acquired from Sigma Aldrich (St. Louis, MO, USA). CellTiter 96^®^ AQueous One Solution Cell Proliferation Assay was obtained from Promega GmbH (Walldorf, Germany). LCMS-grade chloroform, diethyl ether, and hexane were acquired from Fisher Chemical (Pittsburgh, PA, USA). Millipore Sigma™ TLC Plastic Sheets were purchased from Fisher Scientific (Franklin, MA, USA). Penicillin–streptomycin (P-S) was acquired from Bio-Rad (Hercules, CA, USA), and fetal bovine serum (FBS, heat-inactivated, sterile-filtered) was purchased from Lonza (Cologne, Germany). Cisplatin Mylan (1 mg mL^−1^) was acquired from Benta Pharma Industries (Dbayeh, Lebanon), and Ebetaxel (paclitaxel, 6 mg mL^−1^) was purchased from Khalil Fattal & Fils S.A.L. (Beirut, Lebanon), a local agent for Ebewe Pharma GmbH (Unterach am Attersee, Austria). Cell proliferation assay MTS reagent and Phenazine methosulphate (PMS) were acquired from Acros Organics Fisher Scientific (Geel, Belgium).

### 2.2. Extraction and Characterization of CBD

*Cannabis* oil was extracted using ethanol from air-dried Lebanese *Cannabis* flower samples as previously described by Shebaby et al. [26]. The *Cannabis* oil extract (COE) was fractionated by liquid column chromatography using a normal-phase silica gel column and chloroform, hexane, and diethyl ether (4:3:3) as the mobile phase. The fractions were monitored by thin layer chromatography, and similar fractions were combined and concentrated using a rotary evaporator. GC-MS analysis was performed using a Shimadzu GCMS-QP2020NX as previously described by Shebaby et al. [26] (Appendix A).

### 2.3. Cell Proliferation Assays

The anti-proliferative effect of CBD, cisplatin, and paclitaxel on SK-OV-3 and OVCAR-3 cell lines was investigated using the MTS cell proliferation assay. At the day of the experiment, both cell lines were seeded at 5 × 10^5^ cells mL^−1^ in 96-well culture flat bottom plates (100 µL) and left to adhere for 24 h. The different treatment protocols were employed as follows:

### 2.4. Monotherapy

Cells were treated with increasing concentrations of cisplatin (0.62–80 µg mL^−1^), paclitaxel (0.39–25 µg mL^−1^), and CBD (1.56–25 µg mL^−1^) and incubated for 72 h.

### 2.5. Combination Therapy

The cells were treated with varying concentrations of the compounds in six different combinations for 72 h. The dosing ratios of CBD (in DMSO) to chemotherapeutic agent were chosen to be 1:1, 0.5:1, 0.5:0.5, 1.5:1.5, 0.67:1.5, and 0.67:0.67 with respect to their IC_50_ values. To eliminate the possible inactivation of cisplatin by DMSO as reported by Hall et al. [27], the same experiments were repeated on the SK-OV-3 cell line employing ethanol as a solvent.

### 2.6. Priming Assays

A priming experiment was conducted to evaluate the potential of CBD and the chemotherapeutic agents in combination to sensitize the SK-OV-3 cells to the cytotoxic effect of each other. Briefly, 24 h after seeding, cells were treated with the IC_50_ of either agent for 24 h. Then, the medium was discarded, and the wells were washed with DPBS. Cells previously primed with CBD were then treated with increasing concentrations of cisplatin (2.5–80 µg mL^−1^) or paclitaxel (0.39–25 µg mL^−1^) for 48 h, and cells previously primed with cisplatin or paclitaxel were treated with increasing concentrations of CBD (1.56–25 µg mL^−1^) for 48 h as well.

### 2.7. Quantification and Analysis

At the end of the treatment period, MTS/PMS solution (20:1) was added to the cells and incubated in a humidified incubator at 37 °C and 5% CO_2_ atmosphere for three hours. The absorbance intensity of the formazan product was quantified at 490 nm using a Multiskan FC microplate ELISA reader (Thermo Fisher Scientific, Rockford, IL, USA), and the IC_50_ was calculated using GraphPad Prism 9.3.1. As for the combination studies, the type of interaction was analyzed using the median-effect analysis as described by Chou and Talalay (Chou & Talalay, 1984) [28]. The combination indexes (CIs) were calculated using CompuSyn 1.0 software, which allows for the simulation of drug combinations and generation of dose-reduction indices (DRIs). The CI values define the effect in drug combinations as follows: antagonism (CI > 1), additive effect (CI = 1), and synergism (CI < 1).

### 2.8. Statistical Analysis

The data are presented as mean ± standard error mean (SEM) for three or more independent experiments performed in triplicates. Statistical analysis was completed using GraphPad Prism software version 9.3.1. One-way analysis of variance (ANOVA) was used to assess multigroup comparisons, and t-test (with Welch’s correction) was used to evaluate the mean differences between groups. A statistically significant difference was detected at *p* < 0.05. Microsoft Excel 2022 software was used to perform calculations.

## 3. Results

### 3.1. Effect of the Individual Treatment Regimens on Cancer Cell Proliferation

The cytotoxic effects of cisplatin on both ovarian cancer lines OVCAR-3 and SK-OV-3 at 72 h are illustrated in Figure 1A. Cisplatin displayed a significant concentration-dependent cell proliferation inhibitory effect on both cell lines, with IC_50_ values of 1.11 ± 0.24 µg mL^−1^ and 3.30 ± 0.45 µg mL^−1^ of OVCAR-3 and SK-OV-3 cell, respectively. The anti-cancer effect of paclitaxel was also investigated against SK-OV-3 cells only at 72 h (Figure 1B). Paclitaxel demonstrated significant cytotoxicity with an IC_50_ of 9.90 ± 1.06 µg mL^−1^. As for CBD, a significant cytotoxic effect on both cell lines was observed at concentrations of 12.5 µg mL^−1^ and 25 µg mL^−1^ with IC_50_ values of 12.50 ± 1.85 µg mL^−1^ and 12.33 ± 1.35 µg mL^−1^ for OVCAR-3 and SK-OV-3 cell lines, respectively (Figure 1C).

### 3.2. Effect of Combination Treatment Regimens on Cancer Cell Survival

Six different combinations of cisplatin and CBD were selected based on the IC_50_ of the agents and applied on OVCAR-3 and SK-OV-3 cell lines. The mean inhibitory effects (percent of growth inhibition) exerted by each combination was logged in the CompuSyn software for analysis. All effects are reported as percent inhibition. As evident in Table 1, combinations 4 and 5 of cisplatin and CBD exerted the highest growth inhibitory effect on OVCAR-3 cell line, being 66% and 68% inhibition, respectively. However, other combinations exerted suboptimal effect, ranging from 27% to 47% inhibition. Interestingly, combining the IC_50_ of the two compounds resulted in less than 50% inhibition of growth. However, combining half the IC_50_ of cisplatin with the IC_50_ of CBD yielded a comparable effect (47%). This interaction is further characterized by the combination index (CI), which was greater than 1 in all combinations, thus suggesting an antagonistic interaction between cisplatin and CBD in all six regimens when compared to individual treatment. As for the growth inhibitory effect on SK-OV-3 cells, combination 1 (IC_50_ of both drugs) resulted in 50% inhibition of cell growth; combinations 4 and 5 exerted the highest inhibition rate, being 79% and 76% inhibition, respectively (Table 1). The other combinations exhibited less significant inhibition, with the effect ranging between 13% and 40% inhibition rate. Moreover, the CI of all six regimen suggest antagonism between the two drugs (CI > 1) when compared to individual treatment.

Further data were computed using the Chou–Talalay method at different effect levels as indicated by Fa (fraction affected) (Table 2). Fa represents the fractional inhibition associated with every combination index (CI), where a value of Fa = 0.75 signifies a 75% inhibition of cell proliferation. It can be estimated that the combination of cisplatin and CBD displayed a prominent antagonistic interaction through the whole effect range (Fa of 5% to 97% inhibition), with the CI never reaching 1 or lower, suggesting no possibility for synergy or additive effects between the two drugs on OV-CAR-3 cell line. This can be confirmed by the illustrated isolobologram (Figure 2A) where all the data points confirm antagonism once again. The same simulation was applied on the SK-OV-3 cell line, and the results were slightly different (Table 2). Although antagonism was detected (CI > 1), when the inhibition rate reached 95%, another interaction was observed. The CI reaches below 1, CI = 0.89 at 95% inhibition as well as 0.77 at 97% inhibition, suggesting synergism between the two drugs on SK-OV-3 cell line at this high inhibition rate. This phenomenon was not detected with the isobologram, as the highest inhibition level illustrated was 90% (Figure 2B).

Extrapolating current data to actual practice, the drug reduction index (DRI) of each drug is calculated and shown in Table 2. The DRI is a measure of the fold decrease in each drug in a synergistic combination at specific effect levels (Fa) as compared to the individual drug doses when administered independently. DRI > 1 indicates favorable dose reduction, DRI = 1 indicates that there is no need for dose reduction, while DRI < 1 indicates unfavorable dose reduction [28]. The Fa-DRI data in Table 2 are interpreted as follows: to achieve a 95% OVCAR-3 cell growth inhibition, 54.14 µg mL^−1^ of cisplatin is required, and 49.92 µg mL^−1^ of CBD is needed. However, when combining both drugs, to achieve 95% inhibition, it requires 7.04-fold less of cisplatin plus 0.57-fold less (which accounts for a higher dose in this case) of CBD to achieve the same 95% inhibition (i.e., 7.69 µg mL^−1^ of cisplatin and 87.42 µg mL^−1^ of CBD). Similarly for SK-OV-3 cells, the Fa-DRI results (Table 2) extrapolates the data simulated and suggests the following: to achieve a 95% cell growth inhibition, 53.80 µg mL^−1^ of cisplatin is required, and 43.73 µg mL^−1^ of CBD is needed. However, when combining both drugs, to achieve 95% inhibition, it requires 6.30-fold less of cisplatin plus 1.37-fold less of CBD to achieve the same 95% inhibition (i.e., 8.55 µg mL^−1^ of cisplatin and 31.85 µg mL^−1^ of CBD).

To validate the proposed simulation (95% inhibition), an in vitro experiment was performed, applying the simulated concentrations (i.e., 8.9 µg mL^−1^ of cisplatin and 33.18 µg mL^−1^ of CBD) for 72 h on SK-OV-3 cells. The results revealed a 90.3% ± 0.78 inhibition rate. A similar experimental design was carried out for CBD with another chemotherapeutic drug, paclitaxel, on one cell line (SK-OV-3) for proof of concept and test for the resistance-reversal hypothesis. As reported in Table 3, all combinations exerted comparable inhibition rates, ranging from 41% to 58%, with the highest inhibition exerted by combination 4. Interestingly, combining half the IC_50_ of paclitaxel with the IC_50_ of CBD yielded a nearly similar effect (combination 2; 44%) to that of combining the IC_50_ of both (combination 1; 41%). The CI for all combinations was above 1, suggesting antagonism between PTX and CBD on SK-OV-3 cell line when compared to individual treatment.

Further analysis and data simulation, presented in Table 4, revealed that at relevant inhibitory effect level (Fa > 0.9), the CI of combining PTX and CBD on SK-OV-3 cell line was higher than 1, suggesting an antagonistic interaction between the two agents. This can be confirmed visually by analyzing the isobologram in Figure 2C. Further Fa-DRI data in Table 4 suggest the following: to achieve a 95% SK-OV-3 cell growth inhibition, 488.87 µg mL^−1^ of PTX is required, and 43.74 µg mL^−1^ of CBD is needed. However, when combining both drugs, to achieve 95% inhibition, it requires 0.002-fold less of PTX plus 1.19 × 10^−4^-fold less of CBD to achieve the same 95% inhibition, which in turn means a higher concentration is needed of both drugs to achieve the same inhibition rate, which is not relevant and not acceptable in this setting.

To eliminate the effect of the solvent on the growth inhibition rate of the combination treatments, the same experiments were repeated on the SK-OV-3 cell line; however, CBD was dissolved in pure ethanol instead of DMSO. Six combination regimens were implemented as before, based on the IC_50_ of the drugs, and the data analyzed are reported in Appendix A and illustrated in the isobolograms in Appendix A. The obtained results were comparable to those of the DMSO vehicle.

### 3.3. Priming Assay

Before proceeding with the priming assays, the IC_50_ values of CBD, cisplatin, and paclitaxel on SK-OV-3 cell line at 48 h were determined and displayed in Table 5.

#### 3.3.1. Priming with CBD

The cytotoxic effect of cisplatin (Figure 3A) or paclitaxel (Figure 3B) treatment for 48 h on SK-OV-3 cells following priming with the IC_50_ of CBD for 24 h was determined to be concentration-dependent, and the IC_50_ values are presented in Table 5. The IC_50_ of cisplatin post-priming with CBD decreased significantly from 24.74 ± 3.22 µg mL^−1^ to 13.32 ± 4.05 µg mL^−1^, and the IC_50_ of paclitaxel post-priming with CBD decreased markedly from 9.95 ± 0.31 µg mL^−1^ to 6.52 ± 0.82 µg mL^−1^ (Table 5), suggesting a potential role of CBD in sensitizing SK-OV-3 cells to chemotherapy.

#### 3.3.2. Priming with Cisplatin or Paclitaxel

The inhibitory effect of 48 h of CBD treatment on SK-OV-3 cell proliferation following priming with the IC_50_ of either cisplatin (Figure 3C) or paclitaxel (Figure 3D) is shown to be concentration-dependent, and the IC_50_ values are presented in Table 5. The IC_50_ of CBD post-priming with cisplatin decreased significantly from 16.17 ± 0.26 µg mL^−1^ to 11.04 ± 1.98 µg mL^−1^, while the IC_50_ of CBD following priming with paclitaxel decreased significantly from 16.17 ± 0.26 µg mL^−1^ to 11.37 ± 0.58 µg mL^−1^ (Figure 3C,D and Table 5), suggesting a potential role of both cisplatin and paclitaxel in sensitizing SK-OV-3 cells to the effect of CBD.

## 4. Discussion

*Cannabis* has been shown to possess significantly attractive pharmacological benefits in treating various conditions and diseases, such as malaria, pain, rheumatism, lack of appetite, mood disorders, inflammation, emesis, epilepsy, and much more [1,29]. Recently, the anticancer activities of *Cannabis* have become a focus for scientific research [30], due to its potential benefit in minimizing the side effects associated with chemotherapy and combating resistance to therapy. Nonetheless, the extent to which CBD enhances or compromises concurrent chemotherapy treatment is still unclear. The present study is considered one of the few studies to evaluate the anti-cancer effect of CBD, as a monotherapy and in combination, on ovarian cancer cell lines. The cytotoxic effect of CBD, cisplatin and paclitaxel monotherapy was assessed on OVCAR-3 and SK-OV-3 human adenocarcinoma ovarian cancer cell lines. These two cell lines are reported to be resistant to clinically relevant concentrations of cisplatin according to their product data sheets (ATCC, 2021) [31,32]. The results demonstrate that CBD possesses a concentration-dependent growth inhibitory effect on both cell lines. This is in accordance with the findings reported by Fraguas-Sánchez et al. (2020), who studied the effect of CBD on SK-OV-3 ovarian cancer cell line in addition to OAW42 and IGROV-1 cells, and the results showed a similar antiproliferative results in all the studied cell lines [11]. Moreover, while the IC_50_ of cisplatin at 72 h falls in the range reported in the literature [33,34], the data available on paclitaxel are variable and inconsistent without any authors’ explanation about the IC_50_ discrepancies [34,35,36,37]. In our study, the reported IC_50_ was repeatedly confirmed by four independent experiments, each consisting of triplicates.

As for the conducted combination studies, six combinations of varying concentrations of CBD and cisplatin/paclitaxel were selected based on the IC_50_ values of individual agents (Table 1), and the data were analyzed by CompuSyn Software, which is based on the Chou–Talalay method (1984) to simulate data. The Chou–Talalay method, which depends on the median-effect principle (MEP), is derived from the mass-action law, which is based on the enzyme kinetic model and receptor-binding theory through mathematical induction and deduction. This method is validated and has been cited abundantly in scientific journals [13,28,38,39,40,41,42].

Starting with the combination of CBD and cisplatin on SK-OV-3 cell line, the interaction was labeled antagonistic over most inhibitory rates, up until 95% and 97% inhibition is attained, where the first signs of synergism between the two compounds are detected. According to the Chou and Talalay method, synergy at high effect levels (e.g., at Fa > 0.90) is more relevant for its therapeutic (anti-cancer) effect than synergy at low effect levels (e.g., at Fa < 0.3) [28]. This synergism can be further supported by referring to the drug reduction indexes (DRIs) reported in the Fa-DRI results in Table 2, where specific reduced doses of cisplatin and CBD can be predicted to achieve the same level of inhibition. The results suggest that reducing the dose of cisplatin by 6-fold and that of CBD by 1.32-fold, as an example, could theoretically achieve a significant 95% cancer inhibition rate. Extrapolating the results into clinical practice, the patient can theoretically and practically receive lower doses of the toxic chemotherapeutic agent while reaping the same beneficial results, minimizing the side effects associated with them, and overcoming platinum resistance. This simulated data were later confirmed in a separate in vitro experiment, validating the simulation and the method. The antagonism reported in our study on SK-OV-3 cells is consistent with the results reported by Fraguas-Sánchez and colleagues (2020) [11]. As for the OVCAR-3 cell line, pure antagonism over all inhibitory increment levels is observed. On the contrary, a study by Go et al. (2020) reported marked synergistic inhibition of cell viability in head and neck squamous cell carcinoma when combining CBD and cisplatin [12]. Furthermore, synergism was detected when CBD was combined with other chemotherapeutic agents [12,18,43], suggesting a drug- and cancer-cell-type-specific response and interaction.

According to a previous report, DMSO was shown to deactivate cisplatin and other platinum complexes through ligand displacement by replacing one of the chloride ligands or amino moieties of cisplatin, thereby changing the structure of the complex and rendering it inactive or of diminished activity [27]. Therefore, to eliminate the effect of DMSO as a solvent on cisplatin, the same range of experiments were repeated on SK-OV-3 cell line using ethanol as a vehicle for CBD instead of DMSO. The obtained results confirmed the antagonism between CBD and cisplatin in both solvents.

Furthermore, the combination treatment of CBD and paclitaxel on SK-OV-3 demonstrated a significant antagonistic interaction as well, contrary to the findings reported by Fraguas-Sánchez et al. (2020), which suggested the possibility of synergy when combining the two agents [11]. Given the conflicting outcomes and the absence of additional studies addressing this interaction on this cell line, a more comprehensive assessment is warranted to either validate or rebut the current findings. In fact, multiple studies have confirmed the antagonistic behavior of CBD when combined with different chemotherapeutic agents [8,44,45]. Additionally, the chemo-sensitization potential of CBD in SK-OV-3 cells to cisplatin or paclitaxel, and vice versa, was studied in an effort to decrease the concentration needed to achieve the same level of inhibition, and to combat drug resistance. Lower concentrations of chemotherapy, while achieving the same efficacy level, reduce the body’s exposure to the drugs and thus minimize the bothersome side effects associated with them. One approach to optimize therapy involves a sequential administration of the agents, which helps prevent direct antagonistic interactions that were observed with concurrent administration.

In this study, the priming of SK-OV-3 cells with CBD for 24 h before exposure to cisplatin or paclitaxel for 48 h led to a notable decrease in the IC_50_ as compared to the IC_50_ of the chemotherapeutic drugs in the absence of priming. Similarly, the IC_50_ of CBD has decreased significantly following priming with cisplatin or paclitaxel. These results indicate that both CBD and chemotherapy can sensitize cancer cells upon subsequent treatment, improving their efficacy.

Nonetheless, these findings need to be confirmed by an in vivo model to ascertain the effectiveness of priming. A study by Henley and colleagues (2015) demonstrated that priming the breast cancer cell line MDA-MB-231 with CBD for 24 h prior to a 24 h treatment with cisplatin yielded a significant reduction in cell viability, higher than that of either treatment alone or in combination [46]. Whereas normal human mammary cells (MCF10-A) were less sensitive to priming, higher CBD priming concentrations were needed to achieve significant cell viability reduction. It is worth mentioning that the combination of cisplatin and CBD on these cell lines improved their cell viability suggesting antagonism [46]. Similarly, Fraguas-Sánchez et al. (2020) demonstrated that pre-treatment of SK-OV-3 cells with CBD concentrations lower than the IC_50_ resulted in the sensitization of the cells to paclitaxel [11]. Kosgodage and colleagues (2018) suggested that CBD sensitizes cancer cells to chemotherapy through the inhibition of exosomes and microvesicles (EMV) release, which are responsible for tumor spread and the development of chemoresistance [47]. Another study by Nabissi and colleagues (2013) propose that CBD increases TRPV2 expression and activity, increasing chemotherapeutic drug uptake and inducing apoptosis of glioma cells [17]. These studies reiterate the role of CBD in sensitizing cancer cells to the effects of chemotherapy and suggest the potential benefit of sequential, rather than simultaneous, administration of CBD and chemotherapeutic drugs. However, additional research is necessary to optimize the sensitization protocols and to explore whether the effect is specific to certain types of cancer.

The challenge of chemotherapy resistance in treating ovarian cancer presents a significant concern, compounded by the intolerable side effects accompanying existing treatment regimens. Consequently, patients seek alternative treatment options, such as CBD, which is employed as either palliative therapy or in conjunction with conventional chemotherapeutic agents. Additionally, the use of CBD/THC to manage the nausea and vomiting associated with chemotherapy is showing promising results in clinical trials [24,25]. However, the findings of this study call for extreme caution when combining CBD with chemotherapy drugs, as CBD may potentially interfere with the efficacy of these agents, where the observed antagonistic effect appears to be drug- and cancer-cell-line-specific.

## 5. Conclusions

This study explores the anti-cancer activity of CBD, as monotherapy and in combination with two commonly used chemotherapeutic agents, cisplatin and paclitaxel, against platinum-resistant ovarian cancer OVCAR-3 and SK-OV-3 cell lines. CBD monotherapy showed promising and significant tumor growth inhibitory effect against both cell lines. The current study demonstrated that the combination of CBD with cisplatin or paclitaxel displayed diminished inhibition of cell proliferation as compared to individual treatment. CBD and paclitaxel exhibited an antagonistic interaction on all effect levels. In contrast, although antagonism was prominent over most of the inhibitory effect levels with concurrent treatment of CBD and cisplatin, synergy was detected at the highest effect levels, highlighting the potential benefit of combining the two agents at specific concentrations. However, priming with CBD, cisplatin, or paclitaxel has shown significant sensitization of SK-OV-3 cells to the subsequent treatment, stressing the importance of sequential, rather than simultaneous, administration of the drugs. These findings offer valuable insights for optimizing current therapeutic options when integrating CBD with conventional chemotherapy by adopting priming regimens to circumvent the antagonism observed during co-treatment. Further investigations are necessary to validate the current results in vivo and elucidate the mechanism underlying the interaction of CBD with various anticancer drugs.

## Figures and Tables

**Figure 1 biomedicines-13-00520-f001:**
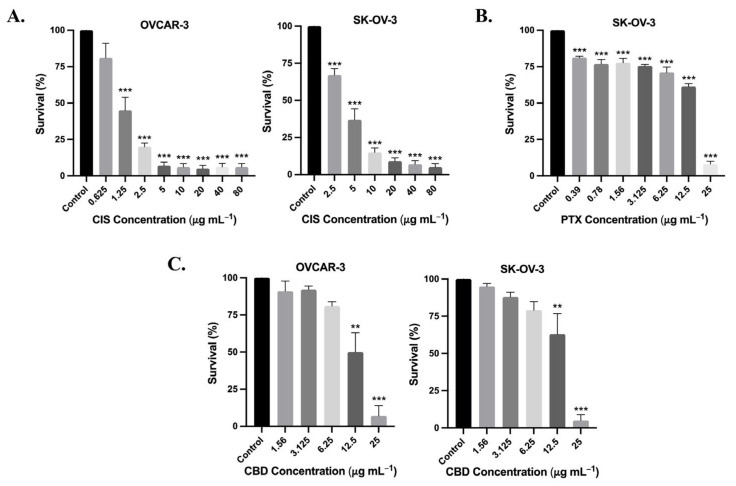
Effect of cisplatin (CIS), paclitaxel (PTX), and CBD on ovarian cancer cell lines. (**A**) OVCAR-3 and SK-OV-3 cells were treated with cisplatin concentrations ranging from 2.5 μg mL^−1^ to 80 μg mL^−1^ using serial dilutions for 72 h. (**B**) SK-OV-3 cells were treated with paclitaxel concentrations ranging from 0.39 μg mL^−1^ to 25 μg mL^−1^ using serial dilutions for 72 h. (**C**) OVCAR-3 and SK-OV-3 cells were treated with CBD concentrations ranging from 1.56 μg mL^−1^ to 25 μg mL^−1^ using serial dilutions for 72 h. Data are expressed as survival rate (%) of cells relative to the control. The mean ± SEM is represented as bar graphs (significant difference detected as compared to the control; ** *p* < 0.01, *** *p* < 0.001).

**Figure 2 biomedicines-13-00520-f002:**
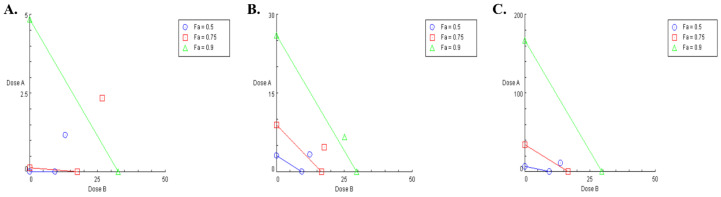
Isobologram for the combination of CBD (in DMSO) with cisplatin on OVCAR-3 (**A**) and SK-OV-3 (**B**) cells (dose A: cisplatin; dose B: CBD) and for the combination of CBD (in DMSO) with PTX on SK-OV-3 cell (**C**) (dose A: PTX; dose B: CBD). Combination data points on the diagonal line indicate an additive effect, while those on the lower left indicate synergism, and those on the upper right indicate antagonism. (Fa: fraction affected).

**Figure 3 biomedicines-13-00520-f003:**
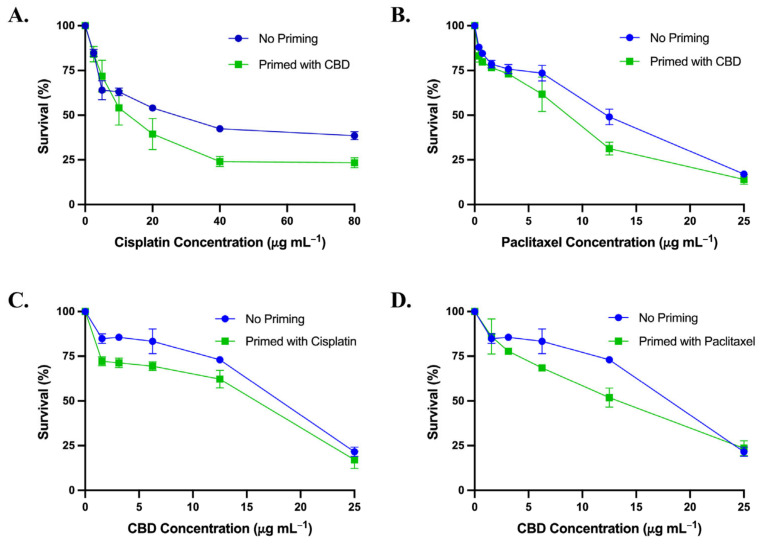
(**A**) Survival curve comparing the cytotoxic effect of cisplatin on SK-OV-3 cell line following priming with CBD vs. no priming. (**B**) Survival curve comparing the cytotoxic effect of paclitaxel on SK-OV-3 cell line following priming with CBD vs. no priming. Survival curves comparing the cytotoxic effect of CBD on SK-OV-3 cell line following priming with cisplatin (**C**) or paclitaxel (**D**) vs. no priming. Data are expressed as survival rate (%) of cells relative to the control.

**Table 1 biomedicines-13-00520-t001:** Inhibitory effect and CI of the six drug combinations of CBD and CIS performed on OVCAR-3 and SK-OV-3 cell lines. (CI: combination index, CIS: cisplatin).

Cell Line	Combination	CIS Concentration (µg mL^−1^)	CBD Concentration (µg mL^−1^)	Effect (% Inhibition)	CI
OVCAR-3	1	1.10	12.5	45	524.02
	2	0.55	12.5	47	202.87
	3	0.55	6.25	27	3392.77
	4	1.65	18.75	66	49.53
	5	0.73	18.75	68	17.21
	6	0.73	8.33	29	3268.91
SK-OV-3	1	3.30	12.30	50	2.38
	2	1.65	12.30	40	2.42
	3	1.65	6.15	13	5.16
	4	4.95	18.45	79	1.43
	5	2.20	18.45	76	1.31
	6	2.20	8.20	15	6.00

**Table 2 biomedicines-13-00520-t002:** CI and DRI data for combination of CBD and CIS on OVCAR-3 and SK-OV-3 cell lines at different levels of fractional inhibition (CI: combination index, DRI: drug reduction index, CIS: cisplatin, Fa: fraction affected).

	OVCAR-3					SK-OV-3				
Fa	CIS Dose (µg mL^−1^)	CBD Dose (µg mL^−1^)	CI	DRI of CIS	DRI of CBD	CIS Dose (µg mL^−1^)	CBD Dose (µg mL^−1^)	CI	DRI of CIS	DRI of CBD
0.05	2.99 × 10^−7^	1.78	587,738	1.70 × 10^−6^	0.89	0.18	1.99	9.76	0.14	0.41
0.10	3.34 × 10^−6^	2.71	85,041.70	1.18 × 10^−5^	0.84	0.37	2.96	6.60	0.22	0.48
0.15	1.49 × 10^−5^	3.52	25,695.10	3.89 × 10^−5^	0.81	0.57	3.77	5.23	0.30	0.53
0.20	4.58 × 10^−5^	4.29	10,435.90	9.58 × 10^−5^	0.79	0.81	4.52	4.43	0.38	0.57
0.25	1.16 × 10^−4^	5.05	4958.58	2.02 × 10^−4^	0.77	1.06	5.26	3.87	0.45	0.60
0.30	2.61 × 10^−4^	5.83	2588.77	3.86 × 10^−4^	0.76	1.36	5.99	3.46	0.53	0.63
0.35	5.45 × 10^−4^	6.63	1434.86	6.98 × 10^−4^	0.75	1.70	6.76	3.12	0.62	0.66
0.40	0.001	7.48	826.33	0.001	0.73	2.09	7.56	2.85	0.71	0.69
0.45	0.002	8.40	487.04	0.002	0.72	2.54	8.42	2.62	0.81	0.72
0.50	0.004	9.42	290.38	0.003	0.71	3.09	9.35	2.41	0.93	0.75
0.55	0.008	10.55	173.37	0.006	0.70	3.76	10.39	2.22	1.06	0.78
0.60	0.01	11.85	102.67	0.01	0.69	4.58	11.57	2.05	1.21	0.82
0.65	0.03	13.37	59.72	0.02	0.68	5.64	12.93	1.89	1.39	0.85
0.70	0.06	15.22	33.77	0.03	0.67	7.03	14.58	1.74	1.61	0.89
0.75	0.14	17.54	18.37	0.06	0.66	8.98	16.63	1.59	1.89	0.94
0.80	0.35	20.65	9.56	0.12	0.64	11.87	19.33	1.44	2.28	0.99
0.85	1.09	25.15	4.85	0.30	0.63	16.64	23.20	1.28	2.86	1.07
0.90	4.85	32.69	2.64	1.02	0.60	26.06	29.57	1.11	3.87	1.18
0.95	54.14	49.92	1.89	7.04	0.57	53.80	43.74	0.89	6.30	1.37
0.97	301.34	67.46	1.86	27.85	0.55	90.12	57.78	0.77	8.90	1.53

**Table 3 biomedicines-13-00520-t003:** Inhibitory effect and combination indexes (CI) of the six drug combinations of CBD and PTX performed on SK-OV-3 cell line. (CI: combination index, PTX: paclitaxel).

Combination	PTX Dose (µg mL^−1^)	CBD Dose (µg mL^−1^)	Effect (% Inhibition)	CI
1	9.90	12.30	41	3.92
2	4.95	12.30	44	2.47
3	4.95	6.15	50	1.35
4	14.85	18.45	58	2.96
5	6.60	18.45	43	3.63
6	6.60	8.20	42	2.48

**Table 4 biomedicines-13-00520-t004:** CI and DRI data for combination of CBD and PTX on SK-OV-3 cell line. (CI: combination index, DRI: drug reduction index, PTX: paclitaxel).

Fa	PTX Dose (µg mL^−1^)	CBD Dose (µg mL^−1^)	CI	DRI of PTX	DRI of CBD
0.05	0.10	1.99	0.004	248.83	3910.63
0.10	0.30	2.96	0.02	54.80	434.84
0.15	0.58	3.77	0.06	21.47	111.62
0.20	0.96	4.52	0.12	10.61	40.09
0.25	1.46	5.26	0.23	5.92	17.21
0.30	2.09	5.99	0.40	3.56	8.22
0.35	2.90	6.76	0.68	2.24	4.20
0.40	3.95	7.56	1.13	1.46	2.24
0.45	5.30	8.42	1.85	0.96	1.23
0.50	7.08	9.35	3.03	0.64	0.68
0.55	9.44	10.39	4.99	0.43	0.38
0.60	12.68	11.57	8.38	0.28	0.21
0.65	17.24	12.93	14.53	0.18	0.11
0.70	23.94	14.58	26.40	0.12	0.06
0.75	34.36	16.63	51.54	0.07	0.03
0.80	51.98	19.33	112.27	0.04	0.01
0.85	85.78	23.20	292.91	0.02	0.004
0.90	166.88	29.57	1070.75	0.007	0.001
0.95	488.87	43.74	9034.59	0.002	1.19 × 10^−4^
0.97	1050.32	57.78	42,001.40	5.61 × 10^−4^	2.49 × 10^−5^

**Table 5 biomedicines-13-00520-t005:** IC_50_ values (µg mL^−1^) of cisplatin, paclitaxel, and CBD on SK-OV-3 cancer cell line following priming at 48 h as compared to no priming.

	IC_50_ (µg mL^−1^)		Significance (*p*-Value)
	Priming with CBD		
	No Priming	Priming	
Treated with CIS	24.74 ± 3.22	13.32 ± 4.05	0.04
Treated with PTX	9.95 ± 0.31	6.52 ± 0.82	0.07
	Priming with CIS		
	No Priming	Priming	
Treated with CBD	16.17 ± 0.26	11.04 ± 1.98	0.04
	Priming with PTX		
	No Priming	Priming	
Treated with CBD	16.17 ± 0.26	11.37 ± 0.58	0.03

## Data Availability

Data are contained within the article and Appendix A.

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
