# Peer review of "Combination of Cannabidiol with Cisplatin or Paclitaxel Analysis Using the Chou–Talalay Method and Chemo-Sensitization Evaluation in Platinum-Resistant Ovarian Cancer Cells"

_biomedicines, 2025, doi:10.3390/biomedicines13020520_

Round 1

Reviewer 1 Report

Comments and Suggestions for Authors

Ismail and coworkers showed the use of Cannabidiol (CBD) with Cisplatin or Paclitaxel in different platinum-resistant ovarian cancer cell lines. This is the first study using these two cell lines in this combination of treatments. The most significant contribution to the field reveals a strategy using CBD to prime cells, leading to their sensitization to the chemotherapeutic agent.

To facilitate the exploitation of the concentration of drugs used in this study given in ug/mL to the most common practice in this field, the authors could include the respective value in uM (micromolar) to the most significant concentrations of the drugs achieved.

One point that remains to be elucidated is that the authors cite the Chou & Talalay method to predict the effects of the combination of therapies. Despite the author's mention that this method was validated and abundantly cited in the literature, only four papers were cited in this study. Could the authors include more examples of the successful use of this application for the same proposal of this study? 

The authors should include a footnote for the abbreviations used in the tables. Please include this information below each table.

Author Response

Title: Combination of Cannabidiol with Cisplatin or Paclitaxel Analysis Using the Chou-Talalay Method and Chemo-Sensitization Evaluation in Platinum-Resistant Ovarian Cancer Cells

Response to Reviewer’s Comments:

Reviewer 1:

Ismail and coworkers showed the use of Cannabidiol (CBD) with Cisplatin or Paclitaxel in different platinum-resistant ovarian cancer cell lines. This is the first study using these two cell lines in this combination of treatments. The most significant contribution to the field reveals a strategy using CBD to prime cells, leading to their sensitization to the chemotherapeutic agent.

We would like to start by thanking the reviewer for taking the time to review the manuscript and provide valuable feedback.

To facilitate the exploitation of the concentration of drugs used in this study given in ug/mL to the most common practice in this field, the authors could include the respective value in uM (micromolar) to the most significant concentrations of the drugs achieved.

Thank you for your suggestion. We can certainly do that. However, we believe that both concentration units is a common practice for in vitro (ng/ml, μg/ml, ..) and in vivo (mg/kg body weight, ..) studies.

One point that remains to be elucidated is that the authors cite the Chou & Talalay method to predict the effects of the combination of therapies. Despite the author's mention that this method was validated and abundantly cited in the literature, only four papers were cited in this study. Could the authors include more examples of the successful use of this application for the same proposal of this study?

Most certainly. We would additionally cite the following studies (highlighted in yellow in main manuscript):

  • Vijayalakshmi, S.; Subramanian, S.; Malathi, S. Chou-Talalay Analysis: Exploring Synergistic Pharmacological Effects of 5-Fluorouracil and Curcumin for Dose Optimization. Int. J. Pharm. Investig. 2024, 14, 843–850, doi:10.5530/ijpi.14.3.94.
  • Duarte, D.; Vale, N. Evaluation of Synergism in Drug Combinations and Reference Models for Future Orientations in Oncology. Curr. Res. Pharmacol. Drug Discov. 2022, 3, 100110, doi:10.1016/j.crphar.2022.100110.
  • Elwakeel, A.; Soudan, H.; Eldoksh, A.; Shalaby, M.; Eldemellawy, M.; Ghareeb, D.; Abouseif, M.; Fayad, A.; Hassan, M.; Saeed, H. Implementation of the Chou-Talalay Method for Studying the in Vitro Pharmacodynamic Interactions of Binary and Ternary Drug Combinations on MDA-MB-231 Triple Negative Breast Cancer Cells. Synergy 2019, 8, 100047, doi:10.1016/j.synres.2019.100047.

The authors should include a footnote for the abbreviations used in the tables. Please include this information below each table.

Thank you for pointing this out. This has been amended.

Reviewer 2 Report

Comments and Suggestions for Authors

In this study, authors have evaluated the anti-cancer activity of cannabidiol (CBD) as a monotherapy and in combination with cisplatin or paclitaxel on human ovarian cancer cells, using the most valid method for this purpose, the Chou-Talalay method and CompuSyn software.

The results are quite varied and complex and sometimes conflict with other studies from other research groups, but in any case they lead the authors to conclude that CBD could improve cancer chemotherapy and that sequential administration of CBD and chemotherapeutic agents is more beneficial than simultaneous administration.

However, this combination is not new and already tested in other cancer types. In addition, some points need to be improved and made clearer to understand.

1.      At the end of the Introduction and again in the Discussion, the authors write that these ovarian cell lines are cisplatin resistant, but from the IC50 values obtained it does not seem so, even in comparison with the data already known from other cell lines. Furthermore, when this is stated, it should be specified in “2. Materials and Methods 2.1. Reagents”, which has not been done. Usually, it’s good practice to indicate whether the resistant phenotype is innate or acquired. In the latter case, it would then be necessary to define the sensitive cell lines from which the resistant counterpart was obtained and with what dose increment and for how long.

Thus, in my opinion, even if the data sheet of these cells reports that are resistant to tumor necrosis factor and to several cytotoxic drugs including diphtheria toxin, cis-platinum and Adriamycin, so I guess innately, for the reasons mentioned above, the reference to platinum-resistance should be modulated in some way, starting from the title.

2.. Figure 1, for a more immediate understanding of the experiment and the data represented, I would add to the title of the concentration axis, also the name of the drug under examination, e.g. cisplatin or paclitaxel concentration (µg mL-1).

 3.      Why paclitaxel was only tested against SK-OV-3 cells and not also against OVCAR-3 cells? Please explain.

4.      In table 1, SK-OV-3 cells appears 2 times but never OVCAR-3 cells. Please correct.

5.      Lines 249-251: Six combination regimens were implemented as before, based on the IC50 of the drugs, and the data analyzed are reported in Tables S1, S2, S3 and S4 and illustrated in the isobolograms in Figures S2 and S3.

But where is this data, I have not been able to find and download it to evaluate it.

6.      Why were the priming experiments conducted on only one of the 2 cell lines? The SK-OV-3 cells? Please explain.

 7.      In the discussion, the authors try to explain the possible reasons why CBD sensitizes cancer cells to chemotherapy referring to previous works by other authors, but there is no speculation regarding the cases in which the combination is antagonistic. Please try to provide possible explanations also for cases where the combination is antagonistic.

Author Response

Title: Combination of Cannabidiol with Cisplatin or Paclitaxel Analysis Using the Chou-Talalay Method and Chemo-Sensitization Evaluation in Platinum-Resistant Ovarian Cancer Cells

Response to Reviewer’s 2 Comments:

In this study, authors have evaluated the anti-cancer activity of cannabidiol (CBD) as a monotherapy and in combination with cisplatin or paclitaxel on human ovarian cancer cells, using the most valid method for this purpose, the Chou-Talalay method and CompuSyn software.

The results are quite varied and complex and sometimes conflict with other studies from other research groups, but in any case, they lead the authors to conclude that CBD could improve cancer chemotherapy and that sequential administration of CBD and chemotherapeutic agents is more beneficial than simultaneous administration.

However, this combination is not new and already tested in other cancer types. In addition, some points need to be improved and made clearer to understand.

Thank you for taking the time to review our manuscript and provide your critical feedback. We are very aware that some results are different from those published from other groups. This realization has led us to repeat our work multiple times than initially planned to confirm our findings, and this is what we did and found. We are confident in our findings as much thorough planning of the study, design, execution and result interpretation has been implemented.

At the end of the Introduction and again in the Discussion, the authors write that these ovarian cell lines are cisplatin resistant, but from the IC50 values obtained it does not seem so, even in comparison with the data already known from other cell lines. Furthermore, when this is stated, it should be specified in “2. Materials and Methods 2.1. Reagents”, which has not been done. Usually, it’s good practice to indicate whether the resistant phenotype is innate or acquired. In the latter case, it would then be necessary to define the sensitive cell lines from which the resistant counterpart was obtained and with what dose increment and for how long.

Thus, in my opinion, even if the data sheet of these cells reports that are resistant to tumor necrosis factor and to several cytotoxic drugs including diphtheria toxin, cis-platinum and Adriamycin, so I guess innately, for the reasons mentioned above, the reference to platinum-resistance should be modulated in some way, starting from the title.

Thank you for your input. We acquired the ovarian cell lines OVCAR-3 from ATCC®, and as the provider specifies in the “Comments” section of the product, this cell line is known to be resistant to clinically relevant concentrations of cisplatin, and we relied on that information and started our study. Please find the details below: https://www.atcc.org/products/htb-161#:~:text=NIH%3AOVCAR-3%20is%20an%20appropriate%20model%20system%20in%20which,be%20useful%20for%20the%20evaluation%20of%20hormonal%20therapy .

We then proceeded to establish the IC50 of cisplatin, and as the available literature is quite variable, we had to establish the IC50 of the cell line that we had on hand, and assess the effects of further combinatorial and priming therapies.

  1. Figure 1, for a more immediate understanding of the experiment and the data represented, I would add to the title of the concentration axis, also the name of the drug under examination, e.g. cisplatin or paclitaxel concentration (µg mL-1).

Well noted. However, the authors believe the figure legends are sufficiently self-explanatory.  

  1. Why paclitaxel was only tested against SK-OV-3 cells and not also against OVCAR-3 cells? Please explain.

This can be answered by the referring to statement mentioned in the Results section (lines 221-223). “A similar experimental design was carried out for CBD with another chemotherapeutic drug, paclitaxel, on one cell line (SK-OV-3) for proof of concept and test for the resistance reversal hypothesis”.

  1. In table 1, SK-OV-3 cells appears 2 times but never OVCAR-3 cells. Please correct.

 Thank you for pointing it out. This has been corrected.

  1. Lines 249-251: Six combination regimens were implemented as before, based on the IC50 of the drugs, and the data analyzed are reported in Tables S1, S2, S3 and S4 and illustrated in the isobolograms in Figures S2 and S3.

But where is this data, I have not been able to find and download it to evaluate it.

Thank you for pointing this out. We will resubmit the Supplementary data again in one file.

  1. Why were the priming experiments conducted on only one of the 2 cell lines? The SK-OV-3 cells? Please explain.

We started the study by choosing two cell lines with acceptably different characteristics that we assumed would behave or respond differently to our intervention. However, as observed by the first part of the project, antagonism between CBD and the chemotherapeutic agents was observed in both cell lines. Thus, we decided to proceed with one cell line only.

  1. In the discussion, the authors try to explain the possible reasons why CBD sensitizes cancer cells to chemotherapy referring to previous works by other authors, but there is no speculation regarding the cases in which the combination is antagonistic. Please try to provide possible explanations also for cases where the combination is antagonistic.

Thank you for your input. We would like to start by referencing the following review paper:

  • Buchtova, T.; Lukac, D.; Skrott, Z.; Chroma, K.; Bartek, J.; Mistrik, M. Drug–Drug Interactions of Cannabidiol with Standard-of-Care Chemotherapeutics. Int. J. Mol. Sci. 2023, 24, 2885, doi:10.3390/ijms24032885.

This work highlights the extensive body of literature available that CBD can act synergistically as well as antagonistically with various chemotherapeutics agents, such as cisplatin and gemcitabine, further supporting our findings. Moreover, we would like to additionally reference the following two studied:

  • Deng, L.; Ng, L.; Ozawa, T.; Stella, N. Quantitative Analyses of Synergistic Responses between Cannabidiol and DNA-Damaging Agents on the Proliferation and Viability of Glioblastoma and Neural Progenitor Cells in Culture. J. Pharmacol. Exp. Ther. 2017, 360, 215–224, doi:10.1124/jpet.116.236968.
  • Huang, T.; Xu, T.; Wang, Y.; Zhou, Y.; Yu, D.; Wang, Z.; He, L.; Chen, Z.; Zhang, Y.; Davidson, D.; et al. Cannabidiol Inhibits Human Glioma by Induction of Lethal Mitophagy through Activating TRPV4. Autophagy 2021, 17, 3592–3606, doi:10.1080/15548627.2021.1885203.

These articles confirm the antagonistic interactions between CBD and other chemotherapeutic agents, including cisplatin.

Round 2

Reviewer 2 Report

Comments and Suggestions for Authors

In this revised version the authors have comprehensively answered the questions raised in the previous revision.